# Position: Evaluation of ML Resource Utilization Requires Model Life Cycle Assessment

**Jared Fernandez** [1]   **Clara Na** [1]   **Yonatan Bisk** [1]   **Constantine Samaras** [2 3]   **Emma Strubell** [1]

## Abstract

Proper accounting of the energy requirements and environmental impact of artificial intelligence (AI) systems is necessary for researchers, developers, policy makers, and users to assess the barriers to building systems at scale. With the growing complexity of pipelines and underlying infrastructure needed to develop and deploy AI systems, previous approaches for evaluating AI efficiency which focus on the costs of a single training run or an individual inference prediction are no longer sufficient. In this position paper, we enunciate **the need for applying life cycle assessment to evaluate the costs of the machine learning model development and deployment pipeline** to properly account for the required resources and downstream impact. Life cycle assessments enable the incorporation of costs across the full life cycle of an AI system and its underlying infrastructure, from the embodied costs associated with the physical computing hardware through the operational costs in training and inference.

## 1. Introduction

As with any emerging technology, an understanding of the resource utilization and byproducts is necessary to both provision the necessary infrastructure and assess their downstream societal impacts. Artificial Intelligence is no different, and its resource requirements remain growing but largely uncertain. The current scaling paradigm, in which large language model performance continues to benefit from increasing scales of computation, has led to growth projections which predict data centers will consume more than 10% of the total U.S. energy demand by 2030 (Green et al., 2024; Shehabi et al., 2024). However, the certainty of such estimates is highly variable and annual load growth expectations vary by more than 4 times (Aljbour et al., 2024).

Reducing the uncertainty in these projections is necessary to ensure that power generation can be properly provisioned while also avoiding placing strain on existing grid infrastructure or increasing utility prices to individual ratepayers (Joint Legislative Audit and Review Commission, 2024). Proper measurement enables informed decision making for governments and industry institutions which have committed hundreds of billions of dollars to investments in computing hardware and energy infrastructure needed to support the development and deployment of large-scale machine learning models with costs rivaling those of "Big Science" projects (OpenAI, 2025b; Bobrowsky, 2025; Isaac, 2025; Smith, 2025; Cai & Sophia, 2025; Parashar et al., 2023).

However, despite the increasing costs of AI and the breadth of stakeholders affected by AI systems, the methodologies we use to evaluate the resource requirements of machine learning models and their socio-economic impacts have not evolved in kind. Existing approaches for evaluating the resource consumption of ML models are often limited to measuring the cost of a marginal step in the development or deployment life cycle of an ML model – i.e. the cost of an individual training run or inference prediction. Measurement of the resource utilization of a system requires aggregation and attribution of those used across all stages in both its production and use, and such approaches focused on the cost of a single constituent stage can fail to account for the effects of efficiency improvements on one stage of a system's life cycle on the resources in another stage.

Fortunately, techniques for analyzing the resource requirements and downstream impacts over the lifetime of manufactured products are well established in the field of industrial ecology; namely, with *life cycle assessment (ISO 14040, ISO 14044 (ISO 14040:2006; ISO 14044:2006))*. Life cycle assessment (LCA) quantifies the impact of a product by decomposing resource consumption and emissions across the stages of manufacture, use, and disposal; and across types of resources (e.g. energy, carbon emissions, human health impacts). The costs associated with previous lifecycle stages,

---
[1]Language Technologies Institute, Carnegie Mellon University, Pittsburgh, PA, USA [2]Department of Civil and Environmental Engineering, Carnegie Mellon University, Pittsburgh, PA, USA [3]Wilton E. Scott Institute for Energy Innovation, Carnegie Mellon University, Pittsburgh, PA, USA. Correspondence to: Jared Fernandez <jaredfern@cmu.edu>.

*Proceedings of the 43rd International Conference on Machine Learning*, Seoul, South Korea. PMLR 306, 2026. Copyright 2026 by the author(s).

such as hardware manufacture and model training, are totaled and amortized through use. Life cycle assessment has been used in semiconductor manufacturing and computing hardware research to quantify the embodied and operational carbon cost of fabrication, recycling, and use of physical hardware (Gupta et al., 2021; Wu et al., 2022; Schneider et al., 2025; Ji et al., 2024; Gupta et al., 2022). However, systematic methods for applying LCA to machine learning models are nascent.

In this paper, **we enunciate the need for life cycle assessment to evaluate the efficiency and environmental impact of machine learning models through development and deployment** by:

1. Presenting the existing landscape for evaluating ML efficiency and resource use, and its limitations (§2)
2. Outlining how these issues can be addressed via application of *life cycle assessment* to machine learning models (§3)
3. Discussing the benefits provided and insights provided by applying LCA to ML models (§5)
4. Providing alternative perspectives on AI's resource requirements (§4)
5. Stating what is needed to enabled life cycle assessment of ML models (§6).

## 2. Limitations in Existing Approaches to Evaluating ML's Resource Needs

In response to the growing resource consumption of machine learning models, there has been a significant increase in scientific inquiry into both (1) the evaluation of ML's resource consumption and environmental impact, and (2) the design of efficient ML methods; as reflected in a myriad of research surveys (Menghani, 2023; Treviso et al., 2023; Tay et al., 2022; Wan et al., 2023; Sui et al., 2025) and publication venues dedicated to the topic (Rezagholizadeh et al., 2024; Dao et al., 2025; Wang et al., 2024b; Sadat Moosavi et al., 2023). Such efforts are necessary first steps towards understanding the overall resource requirements of AI and ML systems. However, existing efforts have often relied on assumptions which are not reflective of the real-world systems and workloads which underpin modern AI systems.

### 2.1. Reliance on Proxy Measures of Efficiency.

A wide range of efficiency metrics have motivated research in the design of efficient machine learning algorithms, model architectures, and computer systems. For example, service-level objectives (SLOs) have been used to optimize cloud serving settings where models are deployed to support latency-sensitive APIs. Whereas the hardware limitations of mobile and edge settings have yielded model compression methods which reduce the memory overheads of models. At the same time, theoretical investigations, which are often

based on proxy metrics for efficiency such as FLOPs, have yielded parameter-, data-, and sample-efficient ML architectures and training algorithms. Although such research yields improvements on these efficiency proxy metrics, such proxies are often not highly correlated with more tangible measures such as latency and energy (Dehghani et al., 2022b; Fernandez et al., 2023). For measures of resource utilization to be informative for stakeholders seeking to standardize or account for resource consumption, reporting must correspond to the real-world quantities of interest.

### 2.2. Failure to Account for Growing Complexity in the ML Model Life Cycle.

Previous efforts to account for resource usage and environmental impacts of machine learning models have mainly focused on the resources consumed for a single stage of the model life cycle – e.g. the energy or water use of large-scale model training (Strubell et al., 2020; Patterson et al., 2021; Faiz et al., 2024; Morrison et al., 2025), the marginal impacts of individual inference predictions (Luccioni et al., 2024; Fernandez et al., 2025a; Patel et al., 2024; Wu et al., 2025; Ding & Shi, 2024; Nguyen et al., 2024), or the embodied costs from manufacture of computing hardware (Li et al., 2025b; 2024b). However, focusing on individual stages of the model life cycle is insufficient to measure the total resources and environmental impact associated with the choice to build a new machine learning model or AI system.

To evaluate the resource demands of AI system, it is necessary to account and attribute the resources consumed across all stages of development and deployment. Conducting such an evaluation is increasingly difficult as the complexity of modern models continues to grow with bespoke deployment and development pipelines. For instance, state-of-the-art large language models (Figure 1), require multiple stages of pre- and post-training, leverage auxiliary models for synthetic data, distillation, and reward modeling; rely on an assortment of inference-time algorithms, and can be deployed across variable hardware platforms. Each stage of the growing model pipeline introduces further complexity to decision-making about development and deployment - as well as additional challenges to accounting of models' resource consumption and environmental impact.

### 2.3. Sector-Wide Projections are Not Grounded in Real Computational Workloads.

Concerns around the rising power demands of AI data centers have led to the rise of various studies that estimate and project growth in data center energy use (Shehabi et al., 2024; Green et al., 2024; Aljbour et al., 2024). To obtain projections on energy use, such studies rely on estimates of future chip shipments and energy efficiency to forecast the

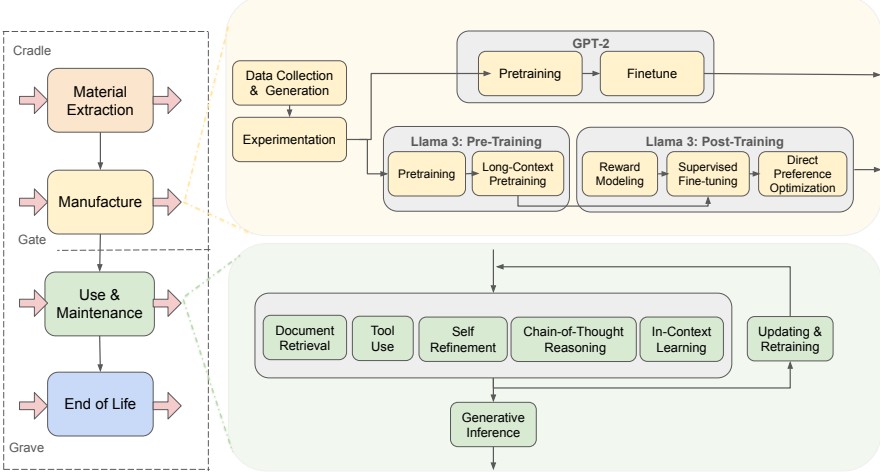

*Figure 1.* LCA enables aggregation across ML model development and deployment life cycles of increasing complexity. The pre- and post-training pipelines of modern LLMs (e.g. OLMo with the Tulu post training recipe Walsh et al. (2025); Lambert et al. (2025)) have significantly more stages than classical train-test settings; and a larger variety of methods for conducting inference (Welleck et al., 2024).

total demands of computing hardware. Sector-level analysis is critical for providing information to developers of electrical grid infrastructure. With infrastructure lead times of multiple years, accurate sector projections enables grid infrastructure to be built out to support the increased capacity demands of data centers, often in excess.

However, these studies rely on assumptions about hardware utilization and energy efficiency at a level of abstraction that obfuscates individual workloads. These assumptions make it impossible to assess for the impact of models developed and deployed by machine learning researchers and practitioners; or to evaluate the impact of model efficiency improvements or design choices.

## 3. Life Cycle Assessment for ML Models

As described in the previous section, existing efforts to measure the resource requirements of ML are limited: often relying on coarse-grained sector-level estimates, failing to measure the real-world resource of interest, or only representing a component of the total costs of an AI system rather than incorporating the total costs across the development and deployment life cycle.

As a means to address these limitations, we believe that **evaluation of the resource demands of ML models requires life cycle assessment**. Life Cycle Assessment (LCA; (Curran, 2006)) provides a methodological basis to determine the environmental and social impacts of a product by accounting for the required resources and environmental impacts of a manufactured product or service through resource extraction, material processing, manufacture, use and disposal (i.e. from *cradle to grave*).

Concretely, LCA's are standardized and defined across two

separate ISO standards. ISO 14040 specifies LCA's conceptual framework, whereas ISO 14044 specified the technical requirements for conducting an LCA (ISO 14040:2006; ISO 14044:2006). [1] For conducting a ML model LCA, we direct practitioners to the practices outlined in ISO 14044.

At the core of LCA is a *functional unit* which defines a quantitative reference for the value provided by a process, which can be compared across potential systems. In turn, a system can be defined that produces the functional unit of interest in relation to resources and emissions.

In this section, we demonstrate how life cycle assessment can be used to enable to more holistic accounting of machine learning models' total resource consumption and environmental impacts for producing a functional unit. We examine the four stages of LCA as defined in ISO standards: *Goal Definition and Scoping*, *Life Cycle Inventory*, *Life Cycle Impact Assessment*, and *Interpretation*.

### 3.1. Goal Definition and Scoping

The first stage of an LCA defines a *functional unit* corresponding to the service being delivered by an AI system and its constraints, and the *system boundaries* which isolate the processes and resource flows encompassed in the study.

#### 3.1.1. FUNCTIONAL UNITS FOR MACHINE LEARNING

Depending on the stakeholder conducting an LCA, the functional unit and the process of interest may vary. For example, institutional developers of large foundation models may be interested in the environmental impact and cost associated

---

[1]Specifically, ISO 14040 specifies the stages of an LCA. By contrast, ISO 14044 provides requirements and guidance on how to conduct an LCA (e.g. considerations for determining system boundaries, factors for assessing the quality of data).

with the development of families of models, and the functional unit could be defined as a "set of trained foundation models for a language task." Downstream users may be concerned with the costs associated with using machine learning models, where a functional unit can be defined as a "processed batch of queries to a machine learning model."

While prior work has performed direct measures of the operational costs of conducting model training and inference, reported values are not comparable when they are not grounded in standardized functional units or consistent system boundaries. For instance, ambiguity in the workloads served render the resulting energy measurements of different model providers, such as OpenAI and Google, difficult to compare (Altman, 2025; Elsworth et al., 2025).

### 3.1.2. PRODUCT SYSTEMS FOR MODERN MACHINE LEARNING MODEL LIFE CYCLES.

After identifying a functional unit for study, a candidate *product system* which produces the functional unit is modeled; defined from: material extraction, manufacture, use and maintenance, through disposal. Input resources and output emissions and waste byproducts are associated with each stage, with *system boundaries* which specify which resource flows to include in the evaluation.

In the case of machine learning models, LCA enables aggregation of costs across the full *product system* encompassing both the resources consumed during hardware manufacture (i.e. *embodied costs*) as well as those consumed during the development and deployment of the model (i.e. *operational costs*). Historically, the process of developing and deploying models followed a simple process of training and performing validation on small sets of in-domain i.i.d. datasets. However, modern ML models are developed using complex pipelines with rapid iterative experimentation processes and multiple stages of training (See Figure 1). Development now encompasses additional stages, such as: neural architecture search, automated machine learning and experimentation, and long-context pretraining, as well as continuous retraining of models during development (Tornede et al., 2023; Sangarya et al., 2024) — each requiring additional resources. Likewise, the variety of methods for model inference has grown, as new paradigms have emerged which shift computation from training to inference to attain higher performance, e.g. via chain-of-thought reasoning, self-refinement, tool use, retrieval-augmented generation, and in-context learning (Welleck et al., 2024).

As with functional units, LCA requires consistency in system boundaries which would otherwise render evaluations incomparable. For instance, differences in the inclusion of Scope 2 offsite water utilization contributed to estimations of water use in by LLMs differing by orders of magnitude (Li et al., 2022; Elsworth et al., 2025).

### 3.2. Life Cycle Inventory

The life cycle inventory stage describes and quantifies the environmental flows associated with the functional unit (National Academies of Sciences, Engineering, and Medicine, 2022). LCA provides an extensible framework which inventories resource and byproduct flows associated with embodied costs (e.g. rare earth minerals, PFAS, CFCs) (Elgamal et al., 2025a;b); as well as those incurred during operational use such as energy, water, carbon emissions and air pollution (Wu et al., 2022; Morrison et al., 2025; Han et al., 2024).

### 3.3. Life Cycle Impact Assessment

Using the quantified costs determined through the life cycle inventory, the total environmental impact of ML models can be determined by translating the inventoried resources into associated impact categories, such as the contribution to global warming from increased emissions; ozone depletion from CFCs; or human health impacts resulting from water depletion, noise, or air quality pollution.

Although LLM developers have begun to report on energy requirements and carbon dioxide emission equivalents (CO2e), the downstream environmental impact remains largely unreported (Dubey et al., 2024; Walsh et al., 2025). Fortunately, life cycle impact assessment provides standard conversions and characterization factors for converting inventoried resources into their associated net environmental impact, such as the U.S. Environmental Protection Agency's Tool for the Reduction and Assessment of Chemical and other environmental Impacts (TRACI) (Bare, 2002; 2011).

### 3.4. Interpretation

For researchers, model developers, and policy makers to utilize the results of an LCA, it is necessary to contextualize and interpret the results of the investigation, by: (1) identifying significant issues with the inventory and assessment; (2) evaluating the completeness, sensitivity, and consistency of data; and (3) providing conclusions and recommendations based on the impact assessment.

Identification of the *significant issues* (i.e. the components of the life cycle that have the greatest impact on the total result) enables location of resource bottlenecks in machine learning models, whether it be the costs associated with hardware fabrication, the upfront costs associated with model training, or the marginal costs of individual inferences. Once the inventory and impact assessment have been validated, the LCA's results enable estimation of the downstream environmental impact of the full machine learning model life cycle– such as to identify which model design choices yield the most efficient system for providing the specified functional unit (e.g., watts per batched inference).

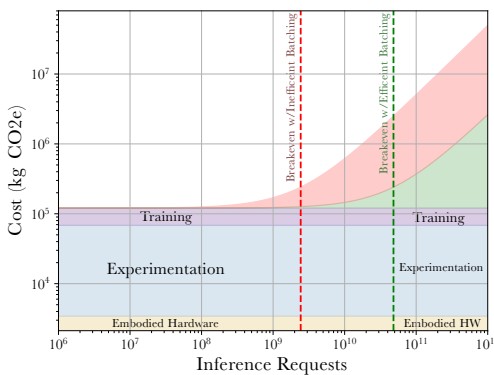

*(a)* **Aggregate costs:** Total environmental impact of models incorporates factors from all stages of model life cycle.[2]Utilizing efficient serving optimizations increases the number of functional units produced under a fixed resource budget.

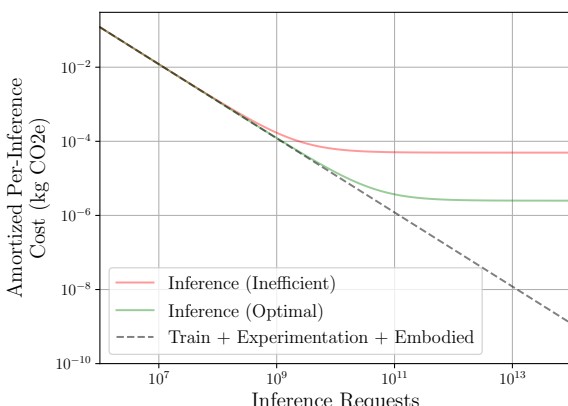

*(b)* **Per-inference costs:** For the functional unit defined as a batch of processed requests, the upfront embodied and operational costs are amortized with use and asymptotically approach the marginal cost of inference.

*Figure 2.* CO2e emissions of OLMo2 7b training and inference (Morrison et al., 2025; Walsh et al., 2025). Increasing inference efficiency via offline batching reduces the unit cost, as does amortization of embodied costs over model use. Decomposition of the resource use across life cycle stages enables identification of the *significant issues* (i.e. the life cycle stage which maximally contributes to total costs).

### 3.5. Case Study: Comparing the Effects of LLM System Design Choices with LCA

As an example, we consider an example LCA with a functional unit corresponding to a batch of processed examples by a large language model (See Figure 2a) with total costs equivalent to the sum of operational inference costs with both upfront training and embodied costs. For our example, computing the cost to produce the functional unit ($C_{\text{FU}}$) requires consideration of not only the marginal cost of inference computation but also the amortized costs of upstream training and hardware manufacturing associated with the inference. More precisely, we consider a simple time-share approach for attributing GPU embodied costs, in which the hardware's manufacturing costs are attributed to training and inference loads based on the length of the workload as fraction of the hardware's usable lifetime. However, we note that the efficacy of different attribution approaches for embodied costs remains open research problems.

$$C_{\text{FU}} = C_{\text{Per Inference}} + \frac{\text{Hardware Utilization Time} \times C_{\text{Embodied}}}{\text{Hardware Lifespan}}$$
$$+ \frac{C_{\text{Experimentation}} + C_{\text{Training}}}{\text{Total Lifetime Inferences}}$$

Accordingly, to compute the cost per functional unit, the practitioner must minimally account and attribute the embodied costs of hardware, operational costs of training, experimentation, and per-request inference. This information can be obtained through direct first-party disclosures from hardware manufacturers (NVIDIA, 2025b;a) for embodied costs; or measured directly in-workload through freely available GPU and CPU utilities (e.g. Nvidia `nvidia-smi` and DCGM; or Intel VTuner) to account for operational cost.

In addition to providing baselines for the total resources required by a machine learning model, LCA can be used to evaluate and provide comparisons across multiple systems that produce the same functional unit, and the relative impact of efficiency improvements to stages of the model life cycle. For modern LLM serving, there exists a variety of design choices that affect the total efficiency and resource consumption, including: parallelization strategies, machine learning software frameworks, cluster scheduling algorithms, and choices in the underlying hardware accelerators. Life cycle assessment enables comparisons of the cost per functional unit when varying such configurations in the context of the full model life time. For instance, shown in Figure 2b, simply increasing the efficiency of LLM serving with increased batching to produce our example functional unit improves hardware utilization, and enables more requests to be served under fixed carbon emissions budgets.

Furthermore, life cycle assessment enables the study of efficiency optimizations that affect multiple stages of the model life cycle. The advent of multi-stage training and inference-time computing yield complex interactions across stages of model development and use which allow for tradeoff of resources between constituent stages. For instance, "reasoning" models (such as DeepSeek-R1, OpenAI GPT-o1, and Gemini) can utilize substantially more resources during inference to attain higher performance on difficult tasks that would otherwise require additional domain-specific training. Alternatively, continual or domain-specific pretraining may extend a model's utility, delaying the need for full model retraining and/or further offsetting the initial training cost.

---

[2]Following Morrison et al. (2025), we ground our inference efficiency estimates in ShareGPT data and likewise assume a 4-year lifetime for GPU hardware.

LCA can enable analysis of the trade-offs of these methods, relative efficiencies, and resource-optimal settings.

Without systematic quantification as part of an LCA, an understanding of the relative magnitudes of efficiency measures across the entire model lifecycle, under different assumptions about the shape of the model lifecycle, remains largely elusive - in Figure 2b, we see that the total per-inference cost is extremely sensitive to the total "lifespan" of the model until it is used at least tens of billions of times.

## 4. Alternative Views

While we believe LCA provides a comprehensive methodology for evaluating the resource requirements and environmental impact of ML models, we acknowledge critiques and alternative views for evaluating AI's resource use.

**Increasing Costs of AI Will Be Offset by Efficiency Improvements in Algorithms, Systems, and Hardware.** Historically, the energy efficiency of computing hardware has doubled approximately every 1.57 years (Koomey et al., 2010). Likewise, the energy efficiency of AI in particular has benefited from additional improvements in software frameworks and model architectures which together are poised to lead to overall reductions in AI's resource demands (Patterson et al., 2022; Oviedo et al., 2025).

However, the increased efficiency of computing hardware has given way to *rebound effects*; such as Jevons' paradox (Jevons, 1866) in which lower cost of use yields increased uptake, leading to increased total resource consumption despite higher utilization and reductions in resources consumed per-unit of resource (Luccioni et al., 2025).

Under the assumption of Jevons' paradox and increased capability and profitability,[3] ML demand will expand to consume all resources that can be allocated to it. In this setting, managing ML's resource consumption becomes less a question of reducing resource use, than one of resource allocation: Given a limited set of resources (e.g. datacenter energy, land), what is the most effective allocation of those resources in order to maximize output? There is a need for methodologies and data enabling analysis of such resource allocations, e.g. between training and inference workloads, across different model types (task-specific, general-purpose), tasks, deployment scenarios, and hardware.

**LCA of Computing Hardware is More Informative than Model-Based LCA.** As datacenters are the physical entities that consume resources and perform the computation of

an ML model, they should be the object of study for LCAs as performed in Gupta et al. (2021); NVIDIA (2025b;a); Schneider et al. (2025). Analysis of the environmental costs associated with hardware provides insight into the efficiency of the underlying computing platform and informs decision-making around hardware provisioning and building of physical infrastructure.

Hardware-based accounting alone does not provide insight into the resource requirements and associated emissions of the *machine learning model* which is often developed and deployed across multiple heterogeneous hardware platforms over the course of its lifetime. For instance, an LCA focused on hardware may examine the carbon emissions associated with the production of computing hardware or the construction of a data center (e.g., the tCO2e attributed to manufacturing a GPU server). While analyses are critical to developers of physical infrastructure, they do not inform model developers and deployers on the resource utilization or environmental impact of the AI system itself. By contrast, model LCAs study the resource requirements associated with the computing workload which utilizes the underlying physical hardware; and provide insight into the tCO2e attributed to serving an LLM request or to train a model.

As opposed to evaluating the resources consumed by the physical infrastructure, the resource consumption of across a model lifecycle is more interpretable to both model developers understanding the costs of their deployment; and consumers, as the model itself is providing the primary functionality of AI systems.

**LCA is Unnecessary as ML Resource Consumption is Concentrated in a Single Life Cycle Stage.** Depending on the use case of an AI system, the total resource requirements can be disproportionately concentrated in a single life cycle stage. For instance, in the case of the largest frontier models, the frequency and scale of inference leads operational inference costs to dominate as the primary contributor to resource consumption – with industry players such as Meta and Google reporting that inference makes up 70% and 60% of their AI power consumption, respectively (Oviedo et al., 2025; Wu et al., 2022; Patterson et al., 2022). In such cases, totaling the marginal costs of inference may be sufficient for determining total costs. Conversely, experimentation and training largely dominate resource consumption for AI researchers as the systems are not deployed at scale and would otherwise require billions of served inferences to amortize the upfront costs of development (Luccioni et al., 2023; Morrison et al., 2025).

Although the impact of fixed embodied and training costs may be amortized through use, the upfront costs of modern ML models remain substantial such that they require deployment at the scale of billions of inference requests before

---

[3]For instance, Microsoft's CEO recently referenced Jevons' paradox to reassure stakeholders that ML efficiency improvements will lead to "skyrocket[ing]" demand for AI, alongside significant investments in energy infrastructure, such as Microsoft's bid to re-commission the nuclear reactor at Three Mile Island in order to power an AI data center.

inference begins to exceed the costs of training (See Figure 2b). LCA additionally provides a robust framework for evaluating emerging applications for AI computation, which may introduce additional stages beyond existing ones and introduce additional computation and resource requirements; such as with multi-agent workflows.

**LCAs are Infeasible Due to Information Unavailability.** LCAs can often rely on information that is disclosed by hardware manufacturers and model providers. As a result, LCA calculations and estimates are often dependent on proprietary information that private corporations may be reluctant to provide to maintain competitive advantage. In such settings, LCAs can still be conducted in the presence of missing data using data from representative averages or secondary sources. As a first step, LCA practitioners and model evaluators could use available disclosures in model tech reports, press releases, and public communications to provide initial approximations of resource utilization. For instance, Google has disclosed aggregate statistics for energy and water use of their Gemini model (Schneider et al., 2025); and OpenAI has disclosed total tokens processed at developer conferences (Ciaccia, 2025). While such statistics may not provide the fine-grained information needed to identify the costs associated with an individual workload, the estimations provided by such a model LCA provide a potential range of resource utilizations for an average or aggregate workload.

That being said, the data challenges for ML model LCAs may be becoming less of an issue as information is increasingly available via: voluntary disclosures (e.g. disclosure of hardware embodied costs for GPUs by NVIDIA (NVIDIA, 2025a;b)), publicly accessible databases (e.g. EcoInvent (Wernet et al., 2016)), and government mandates (e.g. reporting requirements in the EU AI act (European Parliament and Council of the European Union, 2024)). Likewise, conducting direct measurements for ML practitioners is increasingly accessible as many power measurement utilities (e.g., `nvidia-smi`) have existing integrations with commonly used monitoring tools (e.g., `wandb`).

## 5. Benefits of LCAs in Machine Learning

Life cycle assessment enables analysis of the ML model ecosystem for a broad group of stakeholders, including: machine learning researchers and practitioners, users of AI systems, energy and data center providers, policy makers, regulators, and community groups. We discuss insights and decision-making that is enabled should the ML community develop comprehensive LCAs.

**LCA Provides Transparency to Consumers and Enterprise Customers.** Eco-feedback programs, such as the the United States Environmental Protection Agency's Energy Star, have been estimated to reduce consumers' energy use by 5 trillion kWh, saving 4 billion metric tons of CO2e (U.S. Environmental Protection Agency, 2026). While such labels exist for physical appliances, consumers lack information on their individual impact and footprint associated with their use of AI-enabled systems.

LCA of ML models provides a method for quantifying the costs associated with a consumer's use of AI systems, and can enable them to assess the resource footprint associated with their individual choices in AI use. LCAs enable consumers to adjust their utilization to prioritize sustainability, such as by individual reductions in AI usage or opting for less resource intensive models.

**LCA Enables Evaluation of the Effectiveness of Efficiency Research.** Existing evaluations of machine learning models rely on a fragmented assortment of efficiency and task performance benchmarks (Reddi et al., 2020; Mattson et al., 2020; Tschand et al., 2025; Wang et al., 2024a; Yao et al., 2025; Hendrycks et al., 2021). Machine learning algorithms and systems research has addressed efficiency concerns through the development of: model architectures (Wang et al., 2025), efficient serving configurations (Patel et al., 2024; Stojkovic et al., 2025; Shi et al., 2024; Wu et al., 2025; Li et al., 2025b; Stojkovic et al., 2024), improved parallelization algorithms (Hsia et al., 2024; Fernandez et al., 2025b; You et al., 2023; Chung et al., 2024), adaptive inference methods(Li et al., 2024a), power efficient hardware (Patterson et al., 2021; Jouppi et al., 2017), and carbon-aware and demand response (Xing et al., 2023). However, a lack of standardized functional units and treatment of spatio-temporal uncertainty produces research that relies on inconsistent hardware platforms, serving requirements, model architectures, task domains and performance constraints, and metrics for system efficiency – with academic and industry estimates sometimes differing by orders of magnitude (Elsworth et al., 2025).

Life cycle assessment grounds efficiency evaluations with functional units and system boundaries defined in terms of the end-to-end use case, which are more easily compared across systems. As seen in Figure 2b, LCA enables comparative analysis of different system design choices, and determination of which optimizations provided by the research community translate to reductions in real-world resource use and environmental impacts. As such, ML researchers can use LCAs to better identify efficiency bottlenecks across the model life cycle and motivate new lines of research which target lifetime reductions in efficiency. For instance, LCA's could be used to compare resource utilization across stages of training to evaluate whether full retraining is more efficient than alternatives which extend the usable life of a deployed large language model such as continual training which utilizes additional training compute post-deployment or retrieval-augmented generation systems which require additional computation during inference.

**LCA Empowers Developers of AI Models and Power Infrastructure to Effectively Allocate Resources.** Further growth in machine learning is becoming constrained by fundamental limitations in the availability of computing hardware and the energy necessary to power them. Life cycle assessment provides insight into the relative resource consumption and intensity of model training and deployment. By identifying *significant issues* (see §3.4), LCA can be used by machine learning researchers to identify research questions and directions addressing elements of the model life cycle with highest resource consumption and emissions that present the largest bottlenecks to efficiency and opportunities for improvement.

Additionally, LCA enables industry stakeholders to provision and allocate resources so that machine learning systems meet target efficiency and environmental goals — not just in terms of the marginal cost of training or inference, but contextualized within the whole machine learning life cycle. Understanding the relative scales of demand for different life cycle stages enables infrastructure developers to project and accommodate the resources requirements of different workloads (e.g. adapting compute and electrical infrastructure to handle synchronous training or online inference).

**LCA Informs Standards for AI Resource Utilization for Policy Makers.** As the use of AI has grown and energy, water, and other impacts have materialized in many communities, policy makers at federal and local levels are increasingly interested in assessing and mitigating impacts. The energy, carbon, water, air pollutant, noise, and other impacts of AI data centers has driven interest from policy makers and communities for solutions. Lawmakers in multiple countries have introduced bills and passed laws calling for methods to evaluate the resource requirements and sustainability of AI and establish standardized reporting systems (NIST, 2023; 118th United States Congress, 2024; The White House, 2025; European Parliament and Council of the European Union, 2024).

As a commonly used methodology in the assessment of manufactured products in other domains, LCA provides a ready-to-use framework for policy makers and developers of minimum efficiency standards, along with access to LCA practitioners capable of conducting such analyses. Transparently defining scope and functional units can enable guidelines for voluntary or regulated impacts reporting from industry stakeholders, and inform policy maker decisions. For example, the U.S. Inflation Reduction Act (117th United States Congress, 2022) specifies the use of LCA and a model developed by Argonne National Laboratory as the method required to estimate the life cycle greenhouse emissions of hydrogen production to determine eligibility for federal incentives. As policymakers consider both incentives and regulations to minimize the environmental impacts of AI and ML, LCA can enable model comparison and account for both impact and performance. These incentives and regulations can inform industry decision-making regarding model development to account for resource requirements and external impact.

**LCA Improves Accuracy and Completeness in Resource Estimation and Projections.** While the speed and computing requirements of machine learning research have grown with time, the methods required to evaluate the efficiency and resource consumption of the work have not kept up. It is necessary to develop a methodological foundation for grounded assessments of cost.

As shown in Figure 2b, LCA can be used to allocate resource usage to components, and to estimate the relative importance of the constituent stages of hardware fabrication, model training and inference. Additionally, by applying LCA across different types of resources, researchers can account for machine learning's environmental burden along with other key impacts commonly associated with costs of computing such as raw material extraction (Boyd, 2011), water usage(Li et al., 2025a), public health (Han et al., 2024), and per- or polyfluoroalkyl substances (PFAS) (Lee et al., 2025; Elgamal et al., 2025a).

Furthermore, LCA enables researchers to estimate future machine learning systems and provides a tool to understand their potential environmental impact, longterm trends, and rebound effects across a range of scenarios (Luccioni et al., 2025). Evaluating hypothetical systems with differing assumptions enables projection of the impact of: further scaling of ML systems (Hoffmann et al., 2022; Rae et al., 2021), automation of the development process with autoML (Tornede et al., 2023), or alternative hardware platforms such as in edge or mobile settings (Patterson et al., 2024).

## 6. Call to Action: Enabling LCA for ML

Finally, we describe essential components and data gaps to be addressed to enable LCA as a standard practice for analyzing efficiency in machine learning.

**User-Centric Evaluations and Metrics** Variability in the evaluation settings used to characterize efficiency and performance in ML models hinders fair comparison between studies and models. Moreover, while standard efficiency metrics may be measurable and reproducible (e.g. model parameter count, FLOPs), they often fail to map directly to practical user-side requirements such as latency constraints, financial cost, or energy budget (Dehghani et al., 2022a; Fernandez et al., 2023; 2025a). For functional units to correspond to user needs, efficiency benchmarks should not only measure the hardware utilization or speed but be grounded in the performance measured demanded by the use case.

**Transparency in Reporting from Model Developers.** As observed in our example in Figure 2b, the cost of a functional unit of inference is directly dependent on: the serving configurations, hardware selection, and ML system design decisions. Likewise, cost of inference is dependent on the total number of inferences served, a necessary datapoint needed for appropriate attribution of resulting implications to the amortizable training and embodied costs.

While it is increasingly common for developers to release information on the total energy use and estimated carbon emissions of model pretraining, such measurements are often limited to the final training run and fail to account for development costs, the embodied costs of hardware, or the cost and frequency of inference. For downstream users and regulators to accurately assess the cost of ML models, it is necessary for hardware manufacturers and large-scale model developers to release information on the *embodied resources and emissions* associated with hardware fabrication; as well as the *scale, frequency, and settings for model inference*.

Fortunately, there is precedent for both industry-led transparency and joint initiatives between governments and industry model providers. For instance, model providers currently provide self-reporting through the Foundation Model Transparency Index and with Model Cards for frontier model releases (Bommasani et al., 2024; Mitchell et al., 2019). Likewise, both the U.S.'s Center for AI Standards and Innovation and the U.K.'s AI Safety Institute collaborate with model providers OpenAI and Anthropic to conduct pre-release auditing of their model's capabilities (Anthropic, 2025; OpenAI, 2025a; National Institute of Standards and Technology (NIST), 2025). In addition to evaluating for model capabilities, we advocate for a voluntary inclusion of details around the resource requirement and deployment for model LCA in model cards and pre-release audits.

**Standardization of LCA Reporting by Public Institutions.** As industry and private institutions may lack incentives to disclose resource utilization due to competitive advantage and trade secrets, government organizations can work to define LCA-based measurements for AI efficiency evaluations. For instance, both the EU AI Act and the White House's AI Action Plan outline the need for reporting requirements for model providers for general purpose AI (GPAI) models (European Parliament and Council of the European Union, 2024; The White House, 2025). Standards setting organizations and regulators in these regions can define reporting protocols and standards for AI systems, such as the European Commission or European Committee for Standardization; and the National Institute of Standards and Technology or the American National Standards Institute, respectively. Concretely, the EU has already mandated reporting of technical details, including model efficiency, for GPAI models to appropriate government agencies by 2027 (See the EU AI Act Code of Practice: Transparency).

**Improved Granularity in Metrics and Monitoring Tools.** Energy consumption extends beyond the GPU hardware accelerator, including other components of the computing stack such as the CPU, memory, disk, and interconnects (Dodge et al., 2022; McAllister et al., 2024). However, existing reporting is often limited to GPU-only power draw (Dubey et al., 2024; Strubell et al., 2019); or relies on approximations of the energy use upper-bounded by hardware thermal design power (TDP) rather than empirical direct measurements. Furthermore, existing tooling struggles to measure power usage due to processor-to-processor variability and insufficient granularity in sampling frequency (Courty et al., 2024; Verdecchia et al., 2023; Yang et al., 2023). To perform LCA for ML models, direct per-component measurements of resource utilization is needed for all associated computation.

**Increased Research Support and Interdisciplinary Collaboration.** Life cycle assessment is fundamentally interdisciplinary and requires the machine learning research community to pursue collaboration with other fields. In this work, we primarily examine case studies for the energy use and carbon emissions of ML models. However, it is necessary to engage with researchers upstream and downstream of the machine learning model development and deployment ecosystem to understand the full impact in other capacities.

For instance, computer systems experts can provide insight into the efficiency of the underlying computing architectures; and semiconductor researchers can provide insight into the resources required for fabrication and disposal. Additionally, it is necessary to engage with communities directly affected by model use, such as through collaboration with environmental science and public health researchers (Han et al., 2024; Li et al., 2022). Interpretation and action based on results of an LCA is made possible cross-cutting collaborations with domain experts and stakeholders.

Finally, establishing LCA as an accepted practice for use with machine learning models requires the support of and adoption by the research community. Our hope is that machine learning researchers will find nuanced evaluation of practical efficiency and environmental impact to be a compelling and promising research direction.

## Acknowledgments

This work was supported by the National Science Foundation Grant 2326610 and the Graduate Research Fellowship Program under Grant No DGE2140739. Any opinions, findings, conclusions or recommendations expressed in this material are those of the author(s) and do not necessarily reflect the views of the National Science Foundation. This research has been partially supported by Microsoft Corporation as part of the Keio CMU partnership.

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
