# OpenReview forum: "Position: Evaluation of ML Resource Utilization Requires Model Life Cycle Assessment"
_ICML.cc/2026/Position_Paper_Track — ICML 2026 Position Paper Track regular_

### Official Review · Reviewer_HLS2 · 2026-02-14

**Significance:** 2
**Argument Clarity:** 4
**Rating:** 5
**Confidence:** 3

**Questions:**

Regarding the call for transparency, the paper relies heavily on voluntary self-reporting. Given the competitive nature of the industry and the proprietary status of frontier models, how can the proposed LCA framework be effectively implemented if big AI companies refuse to disclose the necessary data?

**Alternative Views Section:**

Yes

**Compliance With Llm Reviewing Policy A Conservative:**

Affirmed.

**Discussion Potential:**

4

**Paper Summary:**

This position paper argues that current methods for evaluating machine learning efficiency—often based on isolated metrics such as training energy consumption or theoretical FLOPs—are insufficient for capturing the full environmental impact of modern AI systems. The authors advocate for the adoption of Life Cycle Assessment (LCA), a “cradle-to-grave” framework that accounts for resource and environmental costs across hardware manufacturing (embodied costs), model training, and deployment (operational costs). Through a case study of an LLM, the paper demonstrates how LCA can reveal important trade-offs, such as the amortization of training and hardware costs over large-scale inference workloads. Based on these insights, the authors advocate for standardized reporting practices and enhanced tooling to facilitate more comprehensive and transparent resource accounting in AI systems.

**Position:**

Yes

**Position In Title:**

Yes

**Related Work:**

4

**Strengths And Weaknesses:**

Strengths:

- The paper is well written and easy to follow.
- The stated position is clearly articulated and represents an important topic worthy of discussion.
- The authors provide multiple pieces of evidence and supporting resources to substantiate the importance of their stated position.
- The authors propose the use of Life Cycle Assessment (LCA), clearly describing its advantages and outlining feasible pathways toward enabling LCA in practice.
- The authors also present alternativeviews, which help provide a more balanced discussion and highlight different viewpoints regarding the feasibility and implications of LCA adoption.

Weaknesses:

- The paper relies on somewhat idealized assumptions regarding the execution of LCA. While the authors correctly identify the need to assess complex pipelines (including pre/post-training, synthetic data generation, and reward modeling), the practical complexity of tracking and auditing these diverse stages is extremely high. The paper does not fully address the operational burden this places on researchers and developers in real-world scenarios.
- The proposed solution relies heavily on transparency from model developers. As the authors acknowledge, many state-of-the-art models are proprietary due to the competitive "AI race". The paper lacks a concrete solution for how to conduct accurate LCAs when essential data (e.g., training duration, hardware scale, manufacturing details) remains a trade secret for the most impactful models.

**Support:**

4

---

> ### Author Rebuttal · Authors · 2026-03-31
>
> We thank the reviewer for their positive feedback and recognition of the importance and comprehensiveness of our arguments. We appreciate the reviewer’s acknowledgement of the importance of standardization and transparency in ML models' resource utilization and the potential insights provided by Life Cycle Assessment for ML models.
>
> Below, we address the reviewer's concerns and weaknesses:
> ### Operational Burden of LCA.
> Although the diversity of lifecycle stages introduces complexity to the development process, we would like to highlight that the necessary tooling for monitoring and evaluating resources across these stages largely already exists. As we discuss in the response to Reviewer ZbC3, measurement utilities for tracking resource utilization are largely publicly available and compatible with existing ML infrastructure (e.g. nvidia-smi and NVIDIA DCGM; and Intel V-Tuner and RAPL for GPU & CPU power measurements, respectively) – with existing integrations in commonly used monitoring infrastructure (e.g. wandb).
>
> As such, the operational burden for the researcher or developer can be simplified to aggregating resource utilization across jobs within each life cycle stage using existing monitoring tools. In addition to providing the requisite information for LCAs, such information is likewise beneficial to developers as it provides insight into the efficiency of their computing systems. Furthermore, we believe that such monitoring does not introduce undue burden to model developers, as there is precedent for large frontier labs accounting for and disclosing the relative costs of experimentation, training, and inference [1].
>
> Additionally, we note that while conducting LCAs introduces additional overhead for auditing ML models, we believe that such evaluations are not at odds with the cost incentives of researchers and model providers. As we discuss in Section 5, identification of efficiency bottlenecks via an LCA can enable effective allocation of computing resources.
>
> ### Conducting LCA’s with Incomplete Data.
> As also discussed in our response to Reviewer p3PS, we agree with the reviewer that the lack of details on industry-scale deployments poses challenges for LCA estimates – and is a challenge that exists for LCAs regardless of domain.
>
> In such settings, LCAs can still be conducted in the presence of missing data using data from representative averages or secondary sources.  As a first step, LCA practitioners and model evaluators could use available disclosures in model tech reports, press releases, and public communications to provide initial approximations of resource utilization. For instance, Google has disclosed aggregate statistics for energy and water use of their Gemini model; and OpenAI has disclosed total tokens processed at recent developer conferences.
>
> Conversely, rather than using aggregate statistics, LCA practitioners evaluating ML models could also calculate resource utilization by directly measuring the resource utilization required by a canonical or representative workload. For instance, industry consortiums like MLCommons have previously developed efficiency benchmarks reflective of canonical training and inference workloads [2,3].
>
> While such statistics may not provide the fine-grained information needed to identify the costs associated with an individual workload, the estimations yielded by such a model LCA nonetheless provide a potential range of the resource utilizations for an average or aggregate workload.
>
> That being said, we would like to highlight that the data challenges for ML model LCAs may be becoming less of an issue as information is increasingly available via: voluntary disclosures (e.g. disclosure of hardware embodied costs for GPU accelerators by NVIDIA [4,5]), publicly accessible databases (e.g. EcoInvent), and government mandates (e.g. reporting requirements in the EU AI act; discussed in more detail in the response to Reviewer FhVN).
>
> ### References
> 1. Wu, Carole-Jean, et al. "Sustainable ai: Environmental implications, challenges and opportunities." Proceedings of machine learning and systems 4 (2022): 795-813.
> 2. Reddi, Vijay Janapa, et al. "Mlperf inference benchmark." 2020 ACM/IEEE 47th Annual International Symposium on Computer Architecture (ISCA). IEEE, 2020.
> 3. Mattson, Peter, et al. "Mlperf training benchmark." Proceedings of Machine Learning and Systems 2 (2020): 336-349.
> 4. NVIDIA. "Product Carbon Footprint (PCF) Summary for NVIDIA HGX H100." 2025, https://images.nvidia.com/aem-dam/Solutions/documents/HGX-H100-PCF-Summary.pdf
> 5. NVIDIA. "Product Carbon Footprint (PCF) Summary for NVIDIA HGX B200." 2025, https://images.nvidia.com/aem-dam/Solutions/documents/HGX-B200-PCF-Summary.pdf

---

> > ### Author Rebuttal · Reviewer_HLS2 · 2026-04-05
> >
> > The authors have addressed most of my concerns in their rebuttal with appropriate clarifications. I will keep the positive score.

---

### Official Review · Reviewer_ZbC3 · 2026-03-11

**Significance:** 3
**Argument Clarity:** 2
**Rating:** 5
**Confidence:** 2

**Questions:**

1. The authors rightfully mention that often only the compute for the last training run is reported which can be a fraction of the compute spent for exploratory experiments. I am missing this information from the example provided in Figure 2a. Is this integrated into the operational training? Furthermore, I think the example would benefit from a more in depth discussions of the details needed for the calculatuon. Are there tools that researches can use to reliably measure energy consumption on large distributed clusters? Additionally, how do you suggest estimating the cost for the embodied hardware in practice? If it's a shared GPU, is this embodied cost amortized across all users/projects? I am sure that going into more detail on the calculations for the presented example would help convince interested readers of the proposal.
2. Are there studies trying to estimate of how much energy requirements/environmental impact stems from pure academic research vs. model development and deployment by private companies?

**Alternative Views Section:**

Yes

**Compliance With Llm Reviewing Policy A Conservative:**

Affirmed.

**Discussion Potential:**

3

**Final Justification:**

My concerns are resolved, provided the authors include the details for the example calculation in the revised manuscript. Explicitly outlining that there is no prior research concerning the environmental impact split between academia and industry would also be a very good addition. Including these updates will make the paper a good contribution and help fuel important discussions in the community.

**Paper Summary:**

The paper argues that machine learning community needs a more rigorous assessment framework of energy requirements and environmental impact of AI models. It outlines the currently fractured landscape of different evaluation techniques, how they vary a lot and fall short of properly taking into account all relevant stages and factors. The authors propose to use the concept of life cycle assessment to be able to estimate the requirements and costs of AI models throughout the development of hardware, as well as the development of the model itself and its inference costs afterwards.

**Position:**

Yes

**Position In Title:**

Yes

**Related Work:**

3

**Strengths And Weaknesses:**

### Is it well supported with reasoning and/or evidence?

The paper's claims are supported by corresponding references. It clearly outlines all of the facets/stages that should be incorporated in the assessment and the factors deciding which stage contributes most an AI model's cost/usage. The authors provide a relevant and easy to follow example in Section 3.5., demonstrating the different pillars of cost as well as how different inference strategies can influence amortized cost of the functional unit. This relatively simple example is helpful and increases the understanding of the proposal for readers who are not familiar with the LCA framework. However, I was missing an explanation of some of the details of the presented example. (see Question 1)

### Is the topic of relevance and importance to the ICML community?

While evaluating the cost of the total lifecycle of ML models is a relevant topic, I have some doubts if the ICML community is the most important target audience. I believe that a large part of conference visitors are people situated within academic research. Therefore it would be interesting and it would strenghten the paper to include numbers/references of how much of current AI costs stem from academic research vs. institutional model development and deployment. (See Question 2)

### Is it likely to inspire discussion?

I believe this is an important conversation to be had and this paper could be the ground for a fruitful discussion of the topic. Also because this is a topic that is not trivial to solve and needs viewpoints from different researchers and industry professionals across all seniority levels.

## Is it clearly argued?

In general, the authors argue towards why their proposal is necessary in order to estimate model's energy requirements. They provide clear arguments with examples of the limitations of the current state, and support their points with respective references. However, when discussing a relevant opposing view (in Section "LCA is Unnecessary as ML Resource Consumption is Concentrated in a Single Life Cycle Stage.") the point the authors want to make is cutoff. (see Line 283, right column)

Furthermore, I noticed some minor mistakes/typos that nevertheless broke the reading flow:
- Line 046, left column: "However, the certainty such estimates is highly variable and..." missing an "of".
- Line 158, left column: "At the core of LCA a functional unit which defines a quantitative reference for the value provided by a process, which can be compared across potential systems." The sentence is missing a verb.
- Subcaption of Figure 2a: The superscript is redirecting to nothing.
- Line 281, left column: "ML demand will expand to consume all resources that can be allocated to it In this setting, managing ML’s resource consumption becomes less a question of reducing resource use, than one of resource allocation:..." is missing a dot.

**Support:**

2

---

> ### Author Rebuttal · Authors · 2026-03-31
>
> We thank the reviewer for their positive assessment of our work and how ML Model Life Cycle Assessment presents a framework for addressing the limitations of existing single-stage metrics.
>
> Below, we address their questions and concerns:
> ## Details of Figure 2.
> We thank the reviewer for their clarification question and acknowledge that Figure 2 can be improved. The figure was intended to separately include *both training and experimentation operational costs* as distinct fixed upfront operational costs, and we have adjusted the figure to include the development stages separately.
>
> ## Evaluating Costs of Experimentation and Embodied Hardware.
> We agree that providing examples of the details and tooling needed for an ML Model LCA can further highlight the immediate actionability of our position. In revision, we will extend our case study in Section 3.5 to also include example methods for conducting the LCA.
>
> Below, we address specific areas of concern:
> * **Required Information**. As the reviewer notes, conducting these calculations requires information, including: embodied hardware costs, training and experimentation costs, expected hardware lifetimes, expected lifetime inferences.
> * **Tools for Energy Measurement**. There exist commonly available tools for evaluating the energy use of both the GPU (e.g. nvidia-smi and DCGM) and CPU (e.g. Intel VTuner). Many of these tools are available in-line with Python libraries compatible with existing ML infrastructure, such as codecarbon and pynvml, or are often logged by default in monitoring utilities such as wandb.
> * **Calculating Experimental, Training, & Embodied Costs**. As the reviewer notes, there are a variety of ways to aggregate and attribute costs. We outline candidate approaches for such calculations:
>     * For costs of training and experimentation, we recommend utilizing existing integrations in wandb for logging system metrics and GPU power draw; and then aggregating costs over experimental runs.
>     * For embodied costs of hardware, we propose utilizing a time-share approach for GPUs as described in Section 3.5 (L246). In this approach, the total embodied costs are amortized over the expected lifetime of the hardware (e.g. 3 to 5 years for GPUs [1] ); and the costs associated with the individual functional unit are allocated as a fraction of the total lifetime hardware costs. The total embodied costs of hardware are increasingly commonly available via hardware LCAs provided by the vendors (e.g., NVIDIA H100 and B200 [2, 3]). We note that the efficacy of different attribution approaches for embodied costs remains open research problems [4].
>
> ## Relevance of LCA to the Academic & ICML Community
> > Are there studies trying to estimate how much of energy requirements/environmental impact stems from pure academic research vs. model development and deployment by private companies?
>
> To the best of our knowledge, there is no prior work comparing the relative costs of academic research and industry deployments from a resource utilization perspective.
>
> > I have some doubts if the ICML community is the most important target audience. I believe that a large part of conference visitors are people situated within academic research.
>
> We note that while the absolute magnitude of resource utilization may be largest in industry settings, the academic research community is often the origin of models and algorithms which that deployed at industry scale.
>
> As we illustrate in Figure 2, the design and selection of efficient methods for ML training and inference can dramatically affect the lifetime efficiency of ML models. We believe that ML model LCAs provide new insight and actionable takeaways for academic researchers, enabling: (1) identification of an ML model’s efficiency bottlenecks (as seen in Figure 2b) which can be used to motivate research directions; and (2) evaluation of the effectiveness of their proposed efficiency interventions (as seen in Figure 2a, and discussed in Section 5: Benefits of LCA, L386).
>
> Furthermore, we believe that standardization of LCAs for ML models is relevant to the academic community, as it serves to create a standardized evaluation framework for analysis of resource utilization – as existing academic and industry estimations of resource utilization can differ by orders of magnitude due to variations in scope and evaluation settings.
>
> Additionally, we appreciate the reviewer’s attention to detail and have gladly corrected the typographic errors in response.
>
> ## References
> 1. Wu, Carole-Jean, et al. "Sustainable ai: Environmental implications, challenges and opportunities." Proceedings of machine learning and systems 4 (2022): 795-813.
> 2. NVIDIA. "Product Carbon Footprint Summary for NVIDIA HGX H100." 2025.
> 3. NVIDIA. "Product Carbon Footprint Summary for NVIDIA HGX B200." 2025.
> 4. Han, Leo, Yueying Lisa Li, and Udit Gupta. "Metrics for Data Center Embodied Carbon." ACM SIGMETRICS Performance Evaluation Review 53.2 (2025): 90-92.

---

> > ### Author Rebuttal · Reviewer_ZbC3 · 2026-04-04
> >
> > Thank you for your answers to my questions and concerns. My concerns are resolved, provided the authors include the details for the example calculation in the revised manuscript. Explicitly outlining that there is no prior research concerning the environmental impact split between academia and industry would also be a very good addition. Including these updates will make the paper a good contribution and help fuel important discussions in the community.

---

### Official Review · Reviewer_p3Ps · 2026-03-12

**Significance:** 4
**Argument Clarity:** 3
**Rating:** 5
**Confidence:** 4

**Questions:**

Minor suggestions

1. Increase font size in Figure 2a, its not quite readable.
2. Fix issues with few overflowing links in the reference.

**Alternative Views Section:**

Yes

**Compliance With Llm Reviewing Policy A Conservative:**

Affirmed.

**Discussion Potential:**

3

**Final Justification:**

This is a good paper, I am in commendation of accepting it.

**Paper Summary:**

This paper argues that evaluating machine learning efficiency -- especially for large language models (LLMs) -- requires a holistic life cycle assessment (LCA) that spans hardware manufacturing, training, experimentation, deployment, and end‑of‑life stages. The authors show that current efficiency metrics (e.g., FLOPs, latency, or per‑training‑run energy) fail to capture the true environmental and societal impacts of modern multi‑stage model pipelines, which now include extensive pretraining, post‑training, continuous updating, and complex inference‑time compute techniques such as chain‑of‑thought reasoning and retrieval. They emphasize the need for standardized functional units, transparent reporting of embodied and operational costs, and consistent system boundaries to enable fair comparisons across models. LCA can reveal the disproportionate contribution of certain stages (e.g., hardware fabrication or training) and guide efficient resource allocation, infrastructure planning, and policy design.

**Position:**

Yes

**Position In Title:**

Yes

**Related Work:**

3

**Strengths And Weaknesses:**

## Strengths

* The paper convincingly argues that existing metrics are inadequate for today’s multi‑stage LLM pipelines and that LCA is the right evaluative lens.
* By showing how amortized per‑inference costs shift with usage and serving efficiency, the paper connects LCA to infrastructure planning, policy, and resource allocation, that are useful for operators and regulators.
* The paper recognizes modern post‑training methods and inference‑time compute (retrieval, CoT, self‑refinement, tool use), strengthening their position.

Overall it's a good paper to read.

## Weakness

* The recommendations hinge on organizations disclosing embodied hardware emissions, development/experimentation energy, and real inference volumes -- data that model vendors and cloud providers seldom publish today, constraining near‑term impact.
* The paper argues LCA can guide policy and resource planning, but provides few empirical case studies that trace LCA‑guided decisions to measurable environmental or operational gains.

**Support:**

3

---

> ### Author Rebuttal · Authors · 2026-03-31
>
> We appreciate the reviewer’s positive assessment of our work and their recognition that LCAs enable a more comprehensive analysis of ML model resource utilization than existing efficiency metrics—which is especially pertinent as AI system complexity increases.
>
> Below, we address the reviewer’s concerns with the practical development and benefits of applying LCAs to ML models.
> ## Information Availability.
> We agree with the reviewer that the lack of details on industry-scale deployments poses challenges for LCA estimates – and is a challenge that exists for LCAs regardless of domain. In fact, we hope that our paper and the development of LCAs for ML models will further bring attention to what information is needed for assessing modern AI systems.
>
> In such settings, LCAs can still be conducted in the presence of missing data using data from representative averages or secondary sources.  As a first step, LCA practitioners and model evaluators could use available disclosures in model tech reports, press releases, and public communications to provide initial approximations of resource utilization. For instance, Google has disclosed aggregate statistics for energy and water use of their Gemini model; and OpenAI has disclosed total tokens processed at developer conferences.
> While such statistics may not provide the fine-grained information needed to identify the costs associated with an individual workload, the estimations provided by such a model LCA provide a potential range of resource utilizations for an average or aggregate workload.
>
> That being said, we would like to highlight that the data challenges for ML model LCAs may be becoming less of an issue as information is increasingly available via: voluntary disclosures (e.g. disclosure of hardware embodied costs for GPU accelerators by NVIDIA [1,2]), publicly accessible databases (e.g. EcoInvent), and government mandates (e.g. reporting requirements in the EU AI act; discussed in more detail in the response to Reviewer FhVN).
>
> ## Guiding Policy Making & Resource Planning.
> In other domains, LCAs can be used to inform selection of raw materials and system design. For instance, in the construction of buildings LCAs are frequently used to assess the impact of utilizing lower carbon alternative materials or the tradeoffs between demolition or refurbishment [3].
>
> For ML models, we believe that LCAs can be used to assess different approaches for designing an AI system that equivalently provides the desired functional unit (i.e. the served LM request in our example in Sec 3.5).
> * **Case Study 1: Updating LLMs for Changing World Knowledge.** LCAs could be used to evaluate different methods for updating an AI system's world knowledge to prevent staleness– such as with comparisons of model retraining, continual retraining, or retrieval augmented generation (RAG) systems which leverage non-parametric contextual information. Each of these approaches introduces additional computation and resource utilization to different stages of the ML model lifecycle. Retraining introduces additional repetitions of the pretraining process, continual pretraining introduces additional training computation post-deployment, and RAG systems often require both additional post-training and inference time computation.
> * **Case Study 2: Domain-Specific vs. General Purpose Models.** LCAs can be used to evaluate whether domain-specific models or general-purpose models are more efficient in total resource utilization for a given task. At the cost of additional training, smaller models fine-tuned for a specific task or domain often achieve comparable performance to general-purpose models that may require more parameters or additional test-time computation. LCAs can provide determine whether the reduced inference costs of domain-specific models offset the additional training costs for fine-tuning.
> To further highlight the impact of LCAs for ML models, we will extend Sec 5 (LCA Enables Evaluation of the Effectiveness of Efficiency Research) to include these examples.
>
> Additionally, thanks for the notes on readability! We will increase the font size for Figure 2 and wrap the urls for camera-ready revision.
>
> ## References
> 1. NVIDIA. "Product Carbon Footprint (PCF) Summary for NVIDIA HGX H100." 2025.
> 2. NVIDIA. "Product Carbon Footprint (PCF) Summary for NVIDIA HGX B200." 2025.
> 3. “Life Cycle Assessment and Buildings.” GSA, U.S. General Services Administration, www.gsa.gov/governmentwide-initiatives/federal-highperformance-buildings/highperformance-building-clearinghouse/integrative-design-strategies/life-cycle-perspective/life-cycle-assessment-and-buildings.
> 4. Luccioni, Sasha, Yacine Jernite, and Emma Strubell. "Power hungry processing: Watts driving the cost of AI deployment?." Proceedings of the 2024 ACM conference on fairness, accountability, and transparency. 2024.

---

> > ### Author Rebuttal · Reviewer_p3Ps · 2026-04-04
> >
> > Thank you for the responses. I will maintain my positive score.

---

### Official Review · Reviewer_FhvN · 2026-03-16

**Significance:** 2
**Argument Clarity:** 2
**Rating:** 4
**Confidence:** 3

**Questions:**

1. How would a Model LCA look in practice?

2. Who would enforce that Model LCAs are issued?

3. What is the difference between the two ISO standards mentioned? How each of them would apply to the idea of a Model LCA?

4. Please address my other comments above.

**Alternative Views Section:**

Yes

**Compliance With Llm Reviewing Policy A Conservative:**

Affirmed.

**Discussion Potential:**

2

**Final Justification:**

Assuming that the authors will explain in more detail in the paper the points that I asked and what they responded to reviewer ZbC3, I do not object acceptance.

**Paper Summary:**

The authors state the position that we should use model life cycle assessment (Model LCA) to evaluate the costs of developing and deploying machine learning models. Essentially, the idea is that we should think about every little aspect and its cost, going from data collection and experimentation to pre-training, post-training, and inference. Moreover, this reporting should be standardized to make sure that it is comparable across the industry, taking as example the ISO standars currently used in industrial ecology.

**Position:**

Yes

**Position In Title:**

Yes

**Related Work:**

3

**Strengths And Weaknesses:**

While I believe that  most of the arguments made in the paper are sound, my main concern is that this paper would not stir debate because it has a very abstract approach. I believe that it would have been more meaningful if the authors showed in concrete terms how a Model LCA would look like, even if for a hypothetical model due to the lack of information for reporting on third-party model. Such a hypotethical Model LCA could then be contrasted with a Hardware LCA, which is described as another possibility in the Alternative Views section. It would have been helpful if there was a discussion of how to enforce that such Model LCAs are issued by model providers and what objections would have been presented by model providers and other stakeholders against such a requirement (rather than what alternatives could accomplish the same goal as Model LCAs). Without those elements, I believe that the paper would not help promoting debate on the topic.

Other comments on the writing:
- Page 1, Section 1, please rewrite "the certainly such estimates is highly variable"
- Page 2, Section 2, correct "resource requirements [of] machine learning models"
- Page 2, first paragraph, correct "However, [the development of] systematic methods"
- Page 3, Section 3, irregular casing in "Life [C]ycle Assessment"
- Page 3, Section 3, correct "be used [to] enable more holistic accounting"
- Page 3, Section 3.1, correct "or pipeline [and] its associated constraints"
- Page 4, Section 3.1, close parentheses in "(i.e., operational costs[)]."
- Page 4, Section 3.1, in "each requires additional resources" use "requiring" instead of "requires"
- Page 4, Footnote 0 should be 1, and the footnote mark is actually shown in Page 5
- Page 6, between two columns, there is clearly something missing in "with industry Meta and Google reporting"
- Page 6, second column, first paragraph ends abruptly with "can often require billions of [what?]"
- Page 9, Anthropic citation is exploding to second column; consider using the xurl package or equivalent

**Support:**

3

---

> ### Author Rebuttal · Authors · 2026-03-31
>
> We acknowledge that Life Cycle Assessments are inherently a *conceptual framework*, as opposed to existing efficiency evaluation benchmarks which rely on different measurements of individual components of the ML model life cycle.  We believe that Life Cycle Assessments for ML models are especially useful as the framework is extensible to models as they grow in complexity.
>
> Additionally, while the LCA framework itself may not be a controversial topic, as the reviewer notes, the exact information requirements, availability and methods of disclosure are a point of contention – which we believe would benefit from further discussion among industry, academic, and policy-making stakeholders.
>
> Below we address the reviewer's comments:
>
> ## Practical Example of a Model LCA
>
> As an example in Section 3, we outline a candidate Model LCA for serving an LLM request. In our example, we examine the OLMo LLM and aggregate resource use over training, experimentation, and inference. We demonstrate how LCA can be used to evaluate the effectiveness of various serving approaches.  To determine the embodied hardware costs and the operational costs, we reference reporting from the Allen Institute for Artificial Intelligence.
>
> To improve actionability and provide guidance on how to compute these values, we will extend Section 3 to include a discussion of the data requirements, methods, and tooling. Due to space limitations, we refer to our response to Reviewer ZbC for additional detail.
>
> ## Contrasting Model and Hardware LCAs.
> Model and Hardware LCAs fundamentally evaluate different objects of study. We will highlight the different results produced and audiences addressed by model and hardware LCAs accordingly in revision.
>
> An LCA focused on hardware may examine the carbon emissions associated with the production of a GPU accelerator or the construction of a data center. For instance, a hardware LCA would provide the tCO2e attributed to manufacturing a GPU server, whereas a model LCA would provide the tCO2e attributed to serving an LLM request. Such analyses are critical to developers of physical infrastructure, but do not inform model developers and deployers on the resource utilization or environmental impact of the *AI system*.
>
> ## Enforcement of LCAs.
> We appreciate the reviewer’s recognition that there are administrative, policy, and incentive challenges to conducting ML Model LCAs. For subsequent revision, we will include these challenges and proposals for enabling LCAs in the Alternative Views and in Section 6: Call to Action.
>
> Both the EU AI Act and the White House’s AI Action Plan outline the need for reporting requirements for model providers for general purpose AI (GPAI) models. Accordingly, we believe that standards setting organizations and regulators in these regions can define reporting protocols and standards for AI systems, such as the European Commission or European Committee for Standardization; and the National Institute of Standards and Technology or the American National Standards Institute, respectively. Concretely, the EU has already mandated reporting of technical details (including the environmental impact of models) for GPAI models to appropriate government agencies by 2027; see the EU AI Act Code of Practice: Transparency and corresponding Model Reporting Form.
>
> The reviewer notes that there may exist industry resistance to providing the necessary information for conducting LCAs. We agree that model providers are resistant to provide information on the scale and use patterns of their models, citing model size and inference load profiles as critical trade secrets. While fine-grained disclosures of exact resource utilization may not be released to the general public, LCAs can nonetheless be conducted using aggregate statistics for usage patterns – some of which are already reported by model providers in academic papers, reports, or in press releases.  Conversely, LCAs can be conducted using directly measurements of the resources required by canonical workloads, which can be determined by consortiums of industry institutions (e.g. MLPerf benchmarks), or by government entities with access to the relevant information.
>
> ## Differences Between ISO 14040 and 14044.
> Together, the two ISO standards specify LCA’s conceptual framework (ISO 14040) and the technical requirements for conducting an LCA (ISO 14044).  Specifically, ISO 14040 specifies the stages of Goal & Scoping, LC Inventory, LC Impact Assessment, and Interpretation; and provides definitions for the components of an LCA (e.g. system boundaries, reference flows). In contrast, ISO 14044 provides requirements and guidance on how to conduct an LCA (e.g. considerations for determining system boundaries, factors to use when assessing the quality of data). For conducting a ML model LCA, we direct practitioners to the requirements and practices outlined in ISO 14044.
>
> Finally, we appreciate the reviewer suggested line-edits and have gladly corrected them.

---

> > ### Author Rebuttal · Reviewer_FhvN · 2026-04-03
> >
> > I appreciate the feedback from the authors. Assuming that the clarifications presented to me and to reviewer ZbC3 are implemented, I am satisfied with the revision.

---

### Decision · Program_Chairs · 2026-04-30

**Decision:**

Accept (regular)

**Comment:**

This position advocates for a broad accounting of cost that may be seen as a unifying framework for many concerns that have been expressed across the ML community, such as energy requirements, carbon footprint, and other "uncaptured" costs associated with models. This could inspire interest among the ICML community both for an accounting of specific interests and (hopefully) serve as a starting point for inspiring discussion about bringing some of these interests together.